# DIFFUSION MODELS AS STRONG ADVERSARIES

## ABSTRACT

Diffusion models have demonstrated their great ability to generate high-quality images for various tasks. With such a strong performance, diffusion models can potentially pose a severe threat to both humans and deep learning models. However, their abilities as adversaries have not been well explored. Among different adversarial scenarios, the no-box adversarial attack is the most practical one, as it assumes that the attacker has no access to the training dataset or the target model. Existing works still require some data from the training dataset, which may not be feasible in real-world scenarios. In this paper, we investigate the adversarial capabilities of diffusion models by conducting no-box attacks solely using data generated by diffusion models. Specifically, our attack method generates a synthetic dataset using diffusion models to train a substitute model. We then employ a classification diffusion model to fine-tune the substitute model, considering model uncertainty and incorporating noise augmentation. Finally, we sample adversarial examples from the diffusion models using the average approximation over the diffusion substitute model with multiple inferences. Extensive experiments on the ImageNet dataset demonstrate that the proposed attack method achieves state-of-the-art performance in both no-box attack and black-box attack scenarios.

## 1 INTRODUCTION

The adversarial vulnerability (Szegedy et al., 2013) of deep learning models is a severe security issue that threatens the deployment of AI applications. Adversarial attacks aim to deceive deep learning models by introducing small perturbations to the input data (Goodfellow et al., 2014; Madry et al., 2017; Dong et al., 2018; Carlini & Wagner, 2017; Croce & Hein, 2020). Based on the knowledge the attacker possesses to generate these perturbations, adversarial attacks can be categorized as white-box attacks or black-box attacks. White-box attacks assume that the attacker has access to the target model's parameters or network structure, allowing them to craft effective adversarial examples. On the other hand, black-box attacks assume that the adversary has no such access and can only interact with the model through input-output queries. Despite this limitation, black-box attacks have shown the ability to achieve high attack success rates against state-of-the-art models in practical scenarios. As a result, the applications of deep learning models face ongoing threats from potential adversaries.

In previous black-box attacks, the transferability of adversarial examples was exploited to deceive the target model. These attacks involved generating adversarial examples against a substitute model that was trained on the same dataset as the target model. However, a more practical scenario is that the adversary may not have access to the training dataset of the target model, where we call this type of attack as a no-box attack (Li et al., 2020). Existing no-box attacks usually require a small amount of data from the training dataset of the target model, which may not be practical in the no-box scenario. Additionally, these attacks require a relatively larger norm perturbation than black-box attacks for successful adversarial examples generation. Therefore, it is still a challenge to conduct effective adversarial attacks under a no-box scenario.

With the advancements in generative models, there is a growing concern regarding their potential threats to humans and deep learning applications. Diffusion models (Ho et al., 2020; Song et al., 2020) are particularly powerful generative models that have gained attention from both the research community and the general public. Open-source text-to-image diffusion models, such as Stable Diffusion (Rombach et al., 2022) and DALL·E 2 (Ramesh et al., 2022), have demonstrated their ability to generate AI-manipulated images that can deceive humans with false information. This raises important security issues that require the attention of the research community in order to address

and mitigate the risks involved. Given the impressive generative capabilities of diffusion models across various tasks, it is worth exploring whether diffusion models can serve as strong adversaries by self-generating training data for no-box attacks. In this paper, we investigate and demonstrate the effectiveness of diffusion models as powerful adversaries under the no-box threat model. We provide a comprehensive discussion of the training process and the attack process involved in leveraging diffusion models for no-box attacks.

Specifically, the training data of the proposed attack is **only** consisting of generated data by the diffusion model for no-box attack. We leverage a technique called classifier-free guidance (Ho & Salimans, 2022) to conditionally generate data using label information from the target model's training dataset, which we utilized the generated data as the training dataset. To provide a comprehensive discussion of the diffusion model, we utilize a classification diffusion model as the substitute model in our attack. This substitute model estimates the distribution of labels based on the input data, employing uncertainty estimation techniques. To improve the transferability of adversarial examples, we introduce scheduled noise during the training of the substitute model. Once the substitute model is trained, we utilize the same diffusion model to generate a dataset for performing no-box unrestricted adversarial attacks. We adopt an ensemble-like approach using the Monte Carlo sampling method over multiple conditional distribution predictions from the diffusion substitute model. We conduct experiments on the ImageNet (Deng et al., 2009) dataset to demonstrate the effectiveness of diffusion models as strong adversaries against deep learning models even in a no-box attack threat model. Our work emphasizes the need for the community to focus on developing more robust defenses against adversarial attacks involving diffusion models.

Our contributions are summarized as follows:

- We propose an effective no-box adversarial attack method using diffusion models against existing deep learning models. Our proposed attack does not rely on any training data or queries from the target model, making it a practical approach for no-box attacks.

- We design effective approaches to generate no-box adversarial examples with diffusion models under the no-box threat model, including the generation method for constructing the dataset, a special fine-tuning method that incorporates model uncertainty and noise augmentation to enhance the model transferability, and a novel ensemble-like no-box unrestricted adversarial attack method that leverages the average prediction from the diffusion substitute model for the generation of strong adversarial examples.

- We conduct extensive experiments to validate the effectiveness of our approach. Our results show that the proposed attack can generate effective no-box and black-box adversarial examples, achieving a state-of-the-art attack success rate compared to existing methods.

## 2 RELATED WORKS

### 2.1 ADVERSARIAL ATTACKS

**White-box attacks.** Szegedy et al. (Szegedy et al., 2013) demonstrated that these models can be vulnerable to imperceptible perturbations, denoted as $x_{\text{adv}} = x + \delta$, which maximize the network's prediction error. The objective of the white-box attack is to find the perturbation that satisfies the constraint $||\delta||_p < \text{dist}$, where $\delta$ represents the perturbation bounded by the $l_p$ norm. In this scenario, the attacker has full knowledge of the target model, including its parameters and network architecture. The perturbations are typically guided by the gradient of the target model's loss function. Adversarial methods such as FGSM (Goodfellow et al., 2014), I-FGSM (Kurakin et al., 2016), and PGD (Madry et al., 2017) are commonly used to perform white-box attacks.

**Black-box attacks.** In a black-box attack scenario, the attacker does not have access to the parameters of the target model and can only make limited queries to the model. Existing black-box attack methods achieve adversarial attacks by leveraging the transferability of a substitute model or estimating the gradient of the target model through multiple queries. However, query-based attacks (Brendel et al., 2017; Chen et al., 2020) typically require a large number of queries to successfully execute a single attack, which may not be feasible in many cases. Recent research efforts have focused on enhancing the adversarial transferability by modifying the backpropagation computation, as seen in approaches like LinBP (Guo et al., 2020), ILA++ (Guo et al., 2022), TAIG (Huang & Kong, 2022), and LGV (Gubri et al., 2022). Another direction is to increase the input diversity to

improve the success rate of black-box attacks. Techniques such as TIM (Dong et al., 2019), SIM (Lin et al., 2019), Admix (Wang et al., 2021), and MBA (Li et al., 2023) have been proposed to achieve this goal. These methods aim to find adversarial examples by exploring different input variations and perturbations.

**No-box attacks.** The no-box threat model, introduced by Li et al. (Li et al., 2020), imposes more practical constraints on the adversary. In this scenario, the attacker is not allowed to access the training data or the outputs of the black-box target model. Only a few correctly labeled data are leaked to the adversary, which limits their knowledge about the target model. Existing works on no-box adversarial attacks leverage the transferability of adversarial examples from a substitute model (Li et al., 2020; Sun et al., 2022). However, these works still rely on using data from the validation set of the target model, which may not be available or permissible in many security-concerned applications. In this paper, we explore the possibility of performing adversarial attacks without any access to the original dataset of the target model. Instead, we rely on the use of synthetic data in our approach. By doing so, we aim to address the limitations of previous works and demonstrate the feasibility of no-box attacks with synthetic data.

## 2.2 DIFFUSION MODELS

Diffusion models have shown great generation quality and diversity in the image synthesis task since Ho et al.(Ho et al., 2020) proposed a probabilistic diffusion model for image generation that greatly improved the performance of diffusion models. Diffusion models for conditional image generation are extensively developed for more usable and flexible image synthesis. Dhariwal & Nichol (Dhariwal & Nichol, 2021) proposed a conditional diffusion model that adopted classifier-guidance for incorporating label information into the diffusion model. They trained the classifier separately and used its gradient for conditional image generation. Jonathan Ho & Tim Salimans (Ho & Salimans, 2022), on the other hand, performed conditional guidance without an extra classifier. They trained a conditional diffusion model alongside a standard diffusion model and used a combination of the two models during sampling. Their idea is motivated by an implicit classifier with the Bayes rule. Followed by (Dhariwal & Nichol, 2021; Ho & Salimans, 2022)'s works, many research (Rombach et al., 2022; Nichol et al., 2021; Lugmayr et al., 2022; Gafni et al., 2022) have been proposed to achieve state-of-the-art performance on image generation, image inpainting, and text-to-image generation tasks. Latent Diffusion Model (LDM) (Rombach et al., 2022) and its text-to-image variant, Stable Diffusion, are capable of generating photo-realistic images. They are able to generate data that is highly related to the dataset of the target model with certain prompts or conditional labels, especially on open-source high-quality datasets like ImageNet (Deng et al., 2009). Besides the data synthesis task, diffusion models achieve satisfying performance on various tasks like classification (Han et al., 2022), segmentation (Baranchuk et al., 2021; Brempong et al., 2022), and representation learning (Kingma et al., 2021; Preechakul et al., 2022). Recent works (Chen et al., 2023a;b; Dai et al., 2023) demonstrated that diffusion models can be used to generate adversarial examples, although these studies have been limited to black-box scenarios and have not thoroughly explored the capabilities of diffusion models as adversaries. Our work will exploit the remarkable generation ability of diffusion models for constructing the training dataset under the no-box threat model.

## 3 PRELIMINARY

### 3.1 DENOISING DIFFUSION IMPLICIT MODELS

Diffusion models, particularly Denoising Diffusion Implicit Models (DDIM), are powerful generative models that can generate high-quality and high-resolution images. The DDIM consists of two main processes: the forward diffusion process and the reverse generation process. The forward diffusion process gradually adds Gaussian noise to the sampled data $\boldsymbol{x}_0$ with the predefined scheduling parameter $\alpha$ and pre-defined $T$ time steps:

$$q_\sigma(\boldsymbol{x}_{t-1}|\boldsymbol{x}_t, \boldsymbol{x}_0) = \mathcal{N}\left(\sqrt{\alpha_{t-1}}\boldsymbol{x}_0 + \sqrt{1 - \alpha_{t-1} - \sigma_t^2} \cdot \frac{\boldsymbol{x}_t - \sqrt{\alpha_t}\boldsymbol{x}_0}{\sqrt{1 - \alpha_t}}, \sigma_t^2\boldsymbol{I}\right) \tag{1}$$

where $q_\sigma(\boldsymbol{x}_T|\boldsymbol{x}_0) = \mathcal{N}(\sqrt{\alpha_T}\boldsymbol{x}_0, (1 - \alpha_T)\boldsymbol{I})$ and $\sigma$ is the magnitude of the Gaussian noise.

The reverse generation process aims to recover the data $\boldsymbol{x}_0$ by a denoising-like process starting with a random noise. With $T$ time steps, we generate a sample $\boldsymbol{x}_{t-1}$ from a sample $\boldsymbol{x}_t$:

$$\boldsymbol{x}_{t-1} = \sqrt{\alpha_{t-1}}\left(\frac{\boldsymbol{x}_t - \sqrt{1-\alpha_t}\epsilon_\theta^{(t)}(\boldsymbol{x}_t)}{\sqrt{\alpha_t}}\right) + \sqrt{1-\alpha_{t-1}-\sigma_t^2}\cdot\epsilon_\theta^{(t)}(\boldsymbol{x}_t) + \sigma_t\epsilon_t \qquad (2)$$

where $\epsilon_t \sim \mathcal{N}(\boldsymbol{0}, \boldsymbol{I})$ is an independent Gaussian noise, and $\epsilon_\theta$ is the trainable model to predict the added Gaussian noise in the forward diffusion process. After training the $\epsilon_\theta$, we will be able to sample high-quality data with a random initial noise.

## 3.2 NO-BOX ADVERSARIAL ATTACK

A no-box adversarial attack is a practical scenario where the adversary does not have access to the target classifier's training dataset or permission to query the model directly. In this case, the attack is carried out by leveraging the transferability of adversarial examples against a substitute model. Previous works have proposed training mechanisms that require only a small amount of data from the target classifier's training dataset. However, in real-world scenarios, it may not be possible to access any data from privacy-concerned applications. In this paper, we propose utilizing the generative capabilities of diffusion models to construct the training dataset for the substitute model. The selection of diffusion models for this purpose needs to meet two requirements: (1) the diffusion models should be open-source and publicly available for practical reasons, and (2) the diffusion models should be capable of generating data that is similar to the training data of the target classifier. To generate the training dataset, we employ conditional labels for DDIM models with classifier-free guidance and prompts with label text for text-to-image diffusion models. By using these techniques, we create a dataset that closely approximates the target classifier's training data. Once the training dataset is obtained, we can train the substitute model using this data to perform the no-box adversarial attack. This allows us to craft adversarial examples that can successfully fool the target classifier, even without direct access to its training data or the ability to query it.

## 3.3 UNRESTRICTED ADVERSARIAL ATTACK WITH DIFFUSION MODELS

The perturbation-based adversarial attack aims to find the smallest possible perturbations to apply to clean data in order to deceive the target classifier. However, with the advent of powerful generative models, these models can directly generate strong adversarial examples as part of their natural generation process. This new form of attack, known as unrestricted adversarial attack (Song et al., 2018), allows for the generation of adversarial examples without relying on small perturbations of clean data.

$$\boldsymbol{x}_{\text{UAE}} = \mathcal{G}(\boldsymbol{z}_{adv}, \boldsymbol{y}), s.t. \boldsymbol{y} \neq f(\boldsymbol{x}_{\text{UAE}}) \qquad (3)$$

where $f(\cdot)$ is the target classifier.

Inspired by Dai's work (Dai et al., 2023), diffusion models are a powerful model to generate human imperceptible unrestricted adversarial examples (UAE). To sample UAEs with the diffusion model, adversarial guidance is added in the reverse generation process:

$$\boldsymbol{x}_{t-1} = \sqrt{\alpha_{t-1}}\left(\frac{\boldsymbol{x}_t - \sqrt{1-\alpha_t}\epsilon_\theta^{(t)}(\boldsymbol{x}_t)}{\sqrt{\alpha_t}}\right) + \sqrt{1-\alpha_{t-1}-\sigma_t^2}\cdot\epsilon_\theta^{(t)}(\boldsymbol{x}_t) + \sigma_t\epsilon_t$$
$$+ a_1 \cdot \sqrt{1-\alpha_t}\nabla_{\boldsymbol{x}_t}\log f(\boldsymbol{y}_a|\boldsymbol{x}_t) \qquad (4)$$

where $a_1$ is the scale of the adversarial guidance and $y_a$ is the target label for the adversarial attack.

The noise sampling guidance is added to the initial noise to better sample the UAEs with prior knowledge:

$$\boldsymbol{x}_T = \boldsymbol{x}_T + a_2 \cdot \sqrt{1-\alpha_T}\nabla_{\boldsymbol{x}_0}\log p_f(\boldsymbol{y}_a|\boldsymbol{x}_0) \qquad (5)$$

where $a_2$ is the noise sampling guidance scale.

## 4 METHODOLOGY

The proposed attack achieves no-box adversarial attacks by UAEs generated by the diffusion models. The substitute model is trained by the generated dataset with the same diffusion models for attack. We will introduce our training mechanisms for the substitute model with the generative ability of diffusion models in Section 4.1, and the fine-tuning method with model uncertainty in Section 4.2. The no-box adversarial attack algorithms will be illustrated in Section 4.3 with detailed discussions.

### 4.1 TRAINING MECHANISMS WITH DIFFUSION MODELS

With the development of diffusion models like the Latent Diffusion Model (LDM) (Rombach et al., 2022) and its successor Stable Diffusion, these models have shown remarkable capabilities in generating high-quality and high-resolution images. Previous works have demonstrated that utilizing generative models as an additional source of training data can enhance the performance of classifiers. However, a crucial question arises: Can we solely rely on generative models for training a classifier? In white-box settings, it is unlikely that the classifier trained solely using generative models will be compatible or optimal. Generative models excel at producing realistic samples, but they may not capture all the complexities and nuances of the real training data that the target classifier has been trained on. However, in real-world scenarios, such as privacy-concerned applications, we cannot always obtain the training details of the target classifier. This practical limitation motivates our approach of using generated data exclusively from diffusion models to train the substitute model. While we acknowledge that relying solely on diffusion models for training may not yield a classifier that matches the target classifier in white-box settings, our objective is to craft effective adversarial attacks given the constraints of the real-world setting, where accessing the target classifier's training details is impractical or prohibited.

Our work considers two training scenarios based on how the diffusion model is trained for a comprehensive discussion on no-box attacks. For standard no-box setting, we adopt pre-trained class-conditional LDM with public checkpoints. The generation of the training dataset is formulated as follows:

$$D \triangleq \{\boldsymbol{x} \sim p(\boldsymbol{x}_T) \prod_{t=1}^{T} p_\theta(\boldsymbol{x}_{t-1}|\boldsymbol{x}_t, \boldsymbol{y})\} \tag{6}$$

where $y$ is the label of the generated data.

A strict no-box scenario is that we assume the diffusion model is trained on multiple datasets without any fine-tuning on the training dataset of the target model. In our work, we use Stable Diffusion 2.0, a text-to-image diffusion model available to the public. To construct the training dataset, we utilize the label text from the target model as prompts for text-to-image generation, which is formulated as:

$$\hat{\epsilon}_\theta^{(t)}(\boldsymbol{x}_t|\boldsymbol{y}) = \epsilon_\theta^{(t)}(\boldsymbol{x}_t|\emptyset) + w \cdot (\epsilon_\theta^{(t)}(\boldsymbol{x}_t|\tau_\theta(\boldsymbol{y})) - \epsilon_\theta^{(t)}(\boldsymbol{x}_t|\emptyset)) \tag{7}$$

where the conditional guidance is incorporated with classifier-free guidance (Ho & Salimans, 2022), and $\tau_\theta(\boldsymbol{y})$ is the text prompt.

After obtaining the training dataset generated using diffusion models, we proceed with the standard training of the substitute model.

### 4.2 FINE-TUNING WITH MODEL UNCERTAINTY

Recent works demonstrate that uncertainty learning is beneficial for the decision-making capabilities of deep learning models. Li et al. (Li et al., 2023) also found that adopting an approximate Bayesian inference technique to the substitute model can enhance the performance of black-box attacks by a large margin. In the case of no-box attacks, it is crucial to avoid overconfident predictions from the substitute model, which may arise due to its under-fitted training on the synthetic dataset generated by diffusion models. To address this, we propose a fine-tuning method that leverages model uncertainty to enhance the transferability of the substitute model.

Diffusion probabilistic models, such as the CARD model proposed by Han et al. (Han et al., 2022), provide an effective way to capture model uncertainty through variational inference. The inference

---

**Algorithm 1** Fine-Tuning Training Algorithm

---

**Require:** $f_\phi$: pre-trained substitute model
**Require:** $\bar{\alpha}$: linear noise schedules for CARD
**Require:** $\hat{\alpha}$: linear noise schedules for LDM
**Require:** $T$: reverse generation process timestep for CARD
**Require:** $T_{\mathrm{LDM}}$: reverse generation process timestep for LDM
 1: **repeat**
 2:     $t_{\mathrm{ft}} \sim \mathrm{Uniform}(\{1 \ldots T_{\mathrm{LDM}}\})$
 3:     Sample $\boldsymbol{x}_{\mathrm{noise}}$ with forward process of the LDM model

$$q(\boldsymbol{x}_{\mathrm{noise}}|\boldsymbol{x}_0) = \mathcal{N}(\boldsymbol{x}_{\mathrm{noise}}; \sqrt{\hat{\alpha}_{t_{\mathrm{ft}}}}\boldsymbol{x}_0, (1 - \hat{\alpha}_{t_{\mathrm{ft}}})\boldsymbol{I})$$

 4:     $y_0 \sim q(\boldsymbol{y}_0|\boldsymbol{x}_{\mathrm{noise}})$
 5:     $t \sim \mathrm{Uniform}(\{1 \ldots T\})$
 6:     $\epsilon \sim \mathcal{N}(\mathbf{0}, \boldsymbol{I})$
 7:     Compute noise estimation loss

$$\mathcal{L}_\epsilon = \left|\left|\epsilon - \epsilon_\theta\big(\boldsymbol{x}_{\mathrm{noise}}, \sqrt{\bar{\alpha}_t}\boldsymbol{y}_0 + \sqrt{1 - \bar{\alpha}_t}\epsilon + (1 - \sqrt{\bar{\alpha}_t})f_\phi(\boldsymbol{x}_{\mathrm{noise}}), f_\phi(\boldsymbol{x}_{\mathrm{noise}}), t\big)\right|\right|^2$$

 8:     Optimization over $\nabla_\theta \mathcal{L}_\epsilon$
 9: **until** Convergence

---

for the classification task is formulated as follows:

$$\boldsymbol{y} \sim p_{\mathrm{CARD}}(\boldsymbol{y}_T) \prod_{t=1}^{T} p_{\mathrm{CARD}\theta}(\boldsymbol{y}_{t-1}|\boldsymbol{y}_t, \boldsymbol{x}) \tag{8}$$

where $\boldsymbol{y}_{t-1} = \gamma_0\hat{\boldsymbol{y}}_0 + \gamma_1\boldsymbol{y}_t + \gamma_2 f_\phi(\boldsymbol{x}) + \sqrt{\tilde{\beta}_t}\epsilon$, and $\boldsymbol{y}_T \sim \mathcal{N}(f_\phi(\boldsymbol{x}), \boldsymbol{I})$. Details are given in the appendix.

Data augmentations like noise injection (Zhou et al., 2019; He et al., 2019) are effective methods to reduce over-fitting and improve the robustness of a deep learning model. As our no-box attack follows the reverse diffusion process to generate adversarial examples, utilizing noise augmentation would further improve the attack performance which makes the substitute model able to classify noisy inputs. Hence, different from simply adding Gaussian noise, our proposed noise augmentation method injects noises from the forward diffusion process. More specifically, the fine-tuning training algorithm is given in Algorithm 1.

The proposed noise augmentation method aids the substitute model for the classification of samples from the reverse diffusion process. Besides, we also include standard geometric transformations to enhance the performance of the substitute model in the initial training.

## 4.3 NO-BOX ADVERSARIAL ATTACKS WITH DIFFUSION MODELS

When conducting adversarial attacks against a classification diffusion model, the goal is to find perturbations that can deceive the model's softmax output, resulting in misclassification. This process is similar to standard adversarial attacks, where the objective is to find small perturbations that can fool the model's decision-making process. Under the no-box attack scenario, it is more practical that we utilize the generative diffusion model that constructs the training data to conduct unrestricted adversarial attacks against the substitute diffusion model. The no-box adversarial attack with diffusion models samples the adversarial examples with the guidance of the gradient from the substitute diffusion model, which is formulated as Equation 4 and 5.

As the classification diffusion model can also be viewed as an approach to model $p(\boldsymbol{y}|\boldsymbol{x}, D)$, we can approximate the exact inference by adopting the Monte Carlo sampling method $p(\boldsymbol{y}|\boldsymbol{x}, D) = \frac{1}{M}\sum_{i=1}^{M} p(\boldsymbol{y}_i|\boldsymbol{x}, D)$, where $p(\boldsymbol{y}_i|\boldsymbol{x}, D)$ is obtained by multiple sampling. The proposed no-box unrestricted adversarial attack is achieved with the ensemble of multiple inferences.

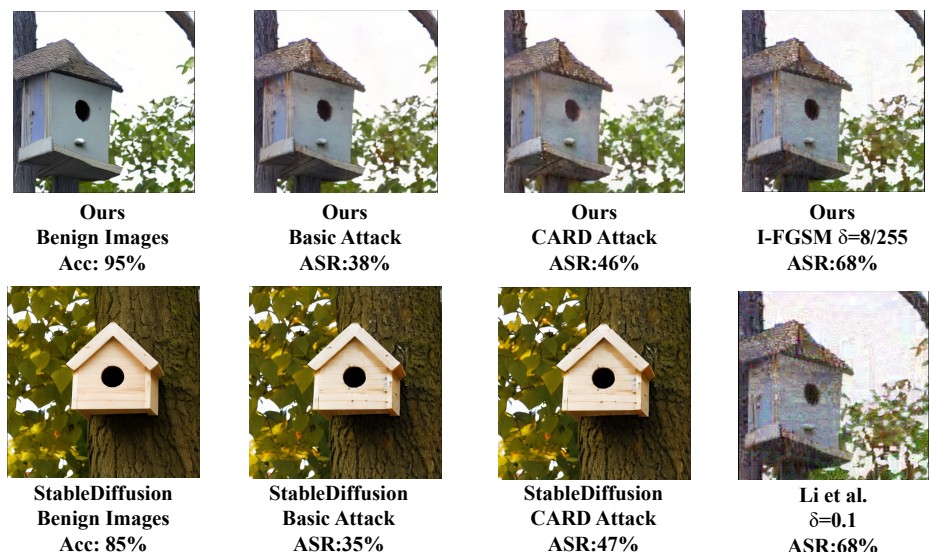

Figure 1: **Comparisons of no-box adversarial examples with our method and Li et al.'s method..**
Note that our method achieves a similar ASR with a significantly lower perturbation.

$$\nabla_{\boldsymbol{x}_t} \log \hat{f}(\boldsymbol{y}_a|\boldsymbol{x}_t) = \frac{1}{M} \sum_{i=1}^{M} \log f(\boldsymbol{y}_a|\boldsymbol{x}_t, \boldsymbol{y}_T, \epsilon_i) \tag{9}$$

where $\epsilon_i \sim \mathcal{N}(0, \boldsymbol{I})$.

## 5 EXPERIMENTS

**Datasets and Substitute Models.** We use the ImageNet (Deng et al., 2009) dataset for major evaluation. Following existing previous work (Li et al., 2023; Guo et al., 2020), we adopt the common settings for transfer-based adversarial attacks. We randomly sample 5000 data from the validation set to conduct the adversarial attack for baselines, and the substitute model is ResNet-50 (He et al., 2016).

**Parameter Settings.** Latent Diffusion Model (LDM) and Stable Diffusion v2.0 (Rombach et al., 2022) are selected as source model in our work. While sampling data, the timestep for the diffusion process is set to 200 for LDM and 50 for Stable Diffusion. Both diffusion models' $\theta$ are set to 0 for deterministic sampling. The classifier-free guidance scale $w$ is set to 3.0 for LDM and 9.0 for Stable Diffusion. We use public checkpoints from the official release for LDM and Stable Diffusion. In fine-tuning with CARD (Han et al., 2022), the diffusion timestep for classification is set to 100. The linear noise schedules are set accordingly as the official implementation. In generating the adversarial examples, we set $N = 5$, $a_1 = 0.75$, $a_2 = 1.0$ for adversarial guidance, and $M = 10$ for multiple inferences.

**Combining with Perturbation-Based Attacks.** Our method aims to generate high-quality non-perturbed adversarial examples with the benign diffusion process. In other words, the sampled adversarial examples can be treated as benign images. Therefore, it is possible to interrogate the perturbation-based adversarial attacks with our generated adversarial examples. Besides using the diffusion model for generating adversarial examples, we perform I-FGSM over the generated examples to enhance their performance on no-box models.

**Target Models.** We select various widely adopted target models for ImageNet to test the attack performance: ResNet-50 (He et al., 2016), VGG-19 (Simonyan & Zisserman, 2014), ResNet-152 (He et al., 2016), Inception v3 (Szegedy et al., 2016), DenseNet-121 (Huang et al., 2017), MobileNet v2 (Sandler et al., 2018), SENet-154 (Hu et al., 2018), ResNeXt-101 (Xie et al., 2017), WRN-101 (Zagoruyko & Komodakis, 2016), PNASNet (Liu et al., 2018), and MNASNet (Tan et al., 2019).

Table 1: Attack success rates of transfer-based no-box attacks on ImagetNet with ResNet-50 as the substitute model, the perturbation of baseline is $\ell_\infty$ with $\delta = 0.1$ and $\delta = 8/255$ for our attack.

| Method | VGG-19 | Inception v3 | ResNet-152 | DenseNet | SENet | WRN | PNASNet | MobileNet | Average |
|---|---|---|---|---|---|---|---|---|---|
| Naïve[†] | 23.80% | 19.14% | 16.24% | 21.06% | 13.00% | 15.84% | 13.04% | 27.56% | 18.71% |
| Prototypical | 80.22% | 63.54% | 62.08% | 70.84% | 55.44% | 62.72% | 51.42% | 82.22% | 66.06% |
| Prototypical* | 81.26% | 66.32% | 65.28% | 73.94% | 57.64% | 66.86% | 54.98% | 83.66% | 68.74% |
| Ours | 52.95% | 30.41% | 36.06% | 39.67% | 29.37% | 40.37% | 26.89% | 50.22% | 38.24% |
| + CARD | 56.67% | 37.40% | 45.93% | 50.07% | 36.37% | 47.87% | 32.53% | 65.33% | 46.52% |
| + CARD I-FGSM | 80.41% | 61.03% | 68.87% | 71.50% | 58.23% | 69.23% | 54.20% | 86.67% | 68.76% |
| $\delta = 0.1$ | **91.60%** | **70.87%** | **77.08%** | **81.60%** | **71.83%** | **77.57%** | **68.38%** | **91.02%** | **78.74%** |
| StableDiffusion | 35.80% | 36.70% | 34.29% | 35.13% | 37.21% | 34.04% | 32.21% | 39.17% | 35.61% |
| + CARD | 46.77% | 45.40% | 46.52% | 46.64% | 48.51% | 49.75% | 38.43% | 51.87% | 46.74% |
| + CARD I-FGSM | 60.71% | 53.84% | 50.12% | 54.51% | 54.32% | 61.99% | 42.04% | 61.03% | 55.19% |
| $\delta = 0.1$ | 63.44% | 58.15% | 56.01% | 60.88% | 55.35% | 70.11% | 47.65% | 69.87% | 60.18% |

Table 2: Attack success rates of transfer-based black-box attacks on ImagetNet with ResNet-50 as the substitute model, the perturbation is $\ell_\infty$ with $\delta = 8/255$.

| Method | ResNet-50 | VGG-19 | ResNet-152 | Inception v3 | DenseNet | MobileNet |
|---|---|---|---|---|---|---|
| I-FGSM | **100.00%** | 39.22% | 29.18% | 15.60% | 35.58% | 37.90% |
| TIM (2019) | **100.00%** | 44.98% | 35.14% | 22.21% | 46.19% | 42.67% |
| SIM (2020) | **100.00%** | 53.30% | 46.80% | 27.04% | 54.16% | 52.54% |
| LinBP (2020) | **100.00%** | 72.00% | 58.62% | 29.98% | 63.70% | 64.08% |
| Admix (2021) | **100.00%** | 57.95% | 45.82% | 23.59% | 52.00% | 55.36% |
| TAIG (2022) | **100.00%** | 54.32% | 45.32% | 28.52% | 53.34% | 55.18% |
| ILA++ (2022) | 99.96% | 74.94% | 69.64% | 41.56% | 71.28% | 71.84% |
| LGV (2022) | **100.00%** | 89.02% | 80.38% | 45.76% | 88.20% | 87.18% |
| MBA (2023) | **100.00%** | 97.79% | 97.13% | 73.12% | 98.02% | 97.49% |
| Ours | **100.00%** | 93.74% | 92.09% | 75.39% | 95.61% | 94.56% |
| Ours + I-FGSM | **100.00%** | 96.03% | 96.01% | **79.24%** | 98.09% | 96.78% |
| Ours + MBA | **100.00%** | **97.93%** | **99.35%** | 76.53% | **98.80%** | **97.71%** |

| Method | SENet | ResNeXt | WRN | PNASNet | MNASNet | Average |
|---|---|---|---|---|---|---|
| I-FGSM | 17.66% | 26.18% | 27.18% | 12.80% | 35.58% | 27.69% |
| TIM (2019) | 22.47% | 32.11% | 33.26% | 21.09% | 39.85% | 34.00% |
| SIM (2020) | 27.04% | 41.28% | 42.66% | 21.74% | 50.36% | 41.69% |
| LinBP (2020) | 41.02% | 51.02% | 54.16% | 29.72% | 62.18% | 52.65% |
| Admix (2021) | 30.28% | 41.94% | 42.78% | 21.91% | 52.32% | 42.40% |
| TAIG (2022) | 24.82% | 38.36% | 42.16% | 17.20% | 54.90% | 41.41% |
| ILA++ (2022) | 53.12% | 65.92% | 65.64% | 44.56% | 70.40% | 62.89% |
| LGV (2022) | 54.82% | 71.22% | 75.14% | 46.50% | 84.58% | 72.28% |
| MBA (2023) | **85.41%** | 94.16% | 95.39% | 77.60% | **97.15%** | 91.33% |
| Ours | 69.60% | 84.50% | 87.92% | 69.88% | 88.93% | 85.22% |
| Ours + I-FGSM | 81.19% | 93.37% | 94.18% | 82.27% | 94.14% | 91.33% |
| Ours + MBA | 84.88% | **97.86%** | **98.55%** | **82.45%** | 95.53% | **92.38%** |

## 5.1 NO-BOX THREAT MODEL

Under the no-box threat model, we compare our method with the state-of-the-art method (Li et al., 2020), where they used supervised ResNets and unsupervised auto-encoders for no-box attacks (Naïve[†] and Prototypical). It should be noted that Li et al.'s method takes 20 images from the original training dataset to train the substitute model, therefore their method only supports attacking a limited number of images and requires a relatively large perturbation under $\delta = 0.1$. Table 1 and Figure 1 show that the sampled images from diffusion models can be correctly classified by the classifier trained on the normal ImageNet dataset, and our attack method can achieve a satisfying ASR under the no-box scenario. Adopting the CARD model can notably improve the transfer ASR by around 10% without increasing the magnitude of the perturbation. Combined with I-FGSM with $\delta = 8/255$, our attack can achieve the state-of-the-art ASR with much smaller perturbations.

## 5.2 BLACK-BOX THREAT MODEL

For a comprehensive discussion on the adversarial ability of diffusion models, we perform standard black-box adversarial attacks with the proposed attack. A variety of state-of-the-art black-box adversarial attacks are selected as comparisons, including LinBP (Guo et al., 2020), ILA++ (Guo et al., 2022), TAIG (Huang & Kong, 2022) and LGV (Gubri et al., 2022), TIM (Dong et al., 2019), SIM (Lin et al., 2019), Admix (Wang et al., 2021) and MBA (Li et al., 2023) with the $\ell_\infty$ attack budget $\epsilon = 8/255$. Table 2 shows that the proposed method is effective at generating transferable adversarial examples. When combined with perturbation-based adversarial attacks, our proposed attack easily outperforms state-of-the-art methods.

Table 3: Attack success rates of transfer-based black-box attacks on ImagetNet against robust models and vision transformers with ResNet-50 as the substitute model, the perturbation is $\ell_\infty$ with $\delta = 8/255$.

| Method | Vision transformers | | | | Robust models | | | |
|---|---|---|---|---|---|---|---|---|
| | ViT-B | DeiT-B | Swin-B | BEiT | Inception v3 | EfficientNet | ResNet-50 | DeiT-S |
| I-FGSM | 4.70% | 5.92% | 5.18% | 3.64% | 11.94% | 9.48% | 9.26% | 10.68% |
| ILA++ (2022) | 9.48% | 21.34% | 14.88% | 11.76% | 15.54% | 30.90% | 10.08% | 11.08% |
| LGV (2022) | 11.18% | 20.02% | 12.14% | 11.66% | 18.00% | 39.06% | 10.56% | 11.50% |
| MBA (2023) | 21.66% | 43.53% | 21.84% | 29.78% | 25.89% | 67.05% | 11.02% | 12.02% |
| Ours | 46.73% | 51.08% | 49.68% | 79.57% | 51.34% | 95.94% | 72.63% | 59.64% |
| Ours + I-FGSM | **52.72%** | **56.87%** | **58.16%** | **81.03%** | **54.38%** | **96.62%** | **75.31%** | **65.26%** |

## 5.3 ADVERSARIAL ROBUST MODELS AND VISION TRANSFORMERS

It has been reported that adversarial defense methods like adversarial training can effectively improve the adversarial robustness of deep learning models. It is practical to test the performance of adversarial attacks under defenses to test the performance on real-world scenarios. We test the performance of various attack methods against adversarial robust models, including adversarial-trained Inception v3, EfficientNet-B0, ResNet-50, and a robust DeiT-S (Touvron et al., 2021).

Vision transformers are recent transformer-based models with state-of-the-art performance but different network artifacts. They achieve relatively high robust accuracy under adversarial attacks for their special feature learning techniques. We also test the attack performance of adversarial examples with recent vision transformers, i.e., ViT-B (Dosovitskiy et al., 2020), a DeiT-B (Touvron et al., 2021), a Swin-B (Liu et al., 2021), and a BEiT (Bao et al., 2021).

Surprisingly, the adversarial examples sampled by diffusion models are better at deceiving the defense methods and vision transformers for its adversarial sampling with the diffusion process rather than adding noise patterns to the image, as shown in Table 3.

## 5.4 DISCUSSION

Experiment results show that even without training data from the target model, our method can achieve state-of-the-art ASR under the no-box threat model. Note that the above 95% benign sampled images from the diffusion model can be correctly classified by the target model. However, if we do not add I-FGSM perturbations to the no-box adversarial examples, the ASR is relatively much lower than the baseline. The reason may be the different data of our proposed attack. Because the data generated by the diffusion model are not from the standard validation set of the training data, they may perform worse on the transferability. Therefore, it is better we use some data from the original training dataset to enhance the attack performance. Furthermore, our adversarial examples exhibit overwhelming performance against the robust model and vision transformers especially combined with existing perturbation-based attacks, which indicates the necessity of designing effective defense mechanisms and through discussion on the defense of adversarial attacks by diffusion models. Diffusion models enable a new and strong type of adversarial attack, which brings new challenges to the improvement of deep learning models' robustness.

## 6 CONCLUSION

In this paper, we investigate the attack ability of diffusion models as strong adversaries. Specifically, we first train the substitute model with the data generated by the diffusion models with label priors from the original training dataset. To further fine-tune the performance of the substitute model, we adopt the classification diffusion probabilistic model to obtain the inference for the classification task. We introduce noise augmentation during the training of the substitute model. This technique helps to reduce over-fitting and enhance the adversarial robustness of the substitute model. After training the substitute model, the adversarial examples are generated by the diffusion model with an ensemble-like attack over the multiple inferences from the classification diffusion substitute model. Extensive experiments on the ImageNet dataset have demonstrated the performance of the proposed attack. We show the strong adversarial ability of diffusion models even without any data or information from the target model. Our work urges effective defense mechanisms against adversarial examples generated by diffusion models.

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
