# DIFFUSION MODELS AS STRONG ADVERSARIES

## A    CARD

The CARD model adopts the output of a pre-trained classifier to infer the label distribution of the input data. Thanks to the characteristics of diffusion models, the CARD model can output different predictions with multiple inferences. By utilizing multiple predictions to estimate the gradient of the pre-trained classifier, the proposed attack can be viewed as an ensemble attack with multiple classifiers. Li et al. (Li et al., 2023) reported that model uncertainty improves the attack transferability by making substitute models more Bayesian, where we found similar results with digging model uncertainty with diffusion classification models.

## B    DETAILED ATTACK SETTINGS FOR DIFFUSION MODEL

As the no-box attack is an extremely challenging attack scenario, we remove the time schedule in the original adversarial guidance, i.e.,

$$\boldsymbol{x}_{t-1} = \sqrt{\alpha_{t-1}} \left( \frac{\boldsymbol{x}_t - \sqrt{1-\alpha_t}\epsilon_\theta^{(t)}(\boldsymbol{x}_t)}{\sqrt{\alpha_t}} \right) + \sqrt{1 - \alpha_{t-1} - \sigma_t^2} \cdot \epsilon_\theta^{(t)}(\boldsymbol{x}_t) + \sigma_t \epsilon_t$$
$$+ a_1 \cdot \nabla_{\boldsymbol{x}_t} \log f(\boldsymbol{y}_a | \boldsymbol{x}_t) \qquad (1)$$

We use fixed $a_1$ when performing the no-box attack, and we clamp the gradient into range $[-0.3, 0.3]$. Additionally, the no-box attack is performed in untargeted attack settings. When performing the black-box attack, we use the same adversarial guidance as (Dai et al., 2023). For the I-FGSM attack, we use 200-step with perturbation $\ell_\infty$ and $\delta = 8/255$. The ASR is calculated with 10 samples from randomly selected 500 classes on an average of 5 attempts.

## C    ETHIC CONCERNS

The motivation for the paper is to completely evaluate the adversarial ability of the diffusion model under the no-box attack scenario. With such a strong generation ability of diffusion models, we show that they are capable of generating adversarial examples without accessing the training dataset of the target model. Additionally, current defense methods all aim at improving the defense ability against perturbation-based attacks. These defenses perform badly against adversarial examples of diffusion models and even worse against the combination of two types of attacks. Therefore, we hope the researchers propose effective defense mechanisms against adversarial diffusion models to enhance the usability and robustness of deep learning models.

## D    WEAKNESS

Despite achieving state-of-the-art performance on the no-box adversarial attack, the proposed attack can generate visually unrealistic images compared to the adversarial examples generated under the black-box scenario. The reason is that adversarial guidance may affect the normal diffusion process if we set $\nabla_{\boldsymbol{x}_t} \log f(\boldsymbol{y}_a | \boldsymbol{x}_t)$ as $-\nabla_{\boldsymbol{x}_t} \log f(\boldsymbol{y} | \boldsymbol{x}_t)$, where $\boldsymbol{y}$ is the ground truth label of the adversarial examples. Additionally, the proposed method that generates the training dataset still has gaps compared to the original dataset. Therefore, the performance of the proposed attack can still need to be improved with better adversarial guidance and dataset generation methods.