# OpenReview forum: "Diffusion Models as Strong Adversaries"
_ICLR.cc/2024/Conference — ICLR 2024 Conference Withdrawn Submission_

### Official Review · Reviewer_BsHh · 2023-10-27

**Soundness:** 4 excellent
**Presentation:** 1 poor
**Contribution:** 2 fair
**Rating:** 5
**Confidence:** 3

**Summary:**

Authors of this paper propose to use diffusion models to craft strong adversarial perturbations against existing deep neural networks in no-box scenario, meaning without access to the training data and without target model queries. The attack is evaluated on the synthetic dataset generated with diffusion models. Few other useful techniques such as model uncertainty and noise augmentation were incorporated to make the attack better.

**Strengths:**

1) Practical setting of adversarial attacks.
2) Both robust convolutional neural networks and vision transformers were evaluated in addition to the classic ResNet, VGGs, DenseNets, MobileNets.
3) Effectiveness of the proposed method in no-box and black-box scenarios.

**Weaknesses:**

1) In my opinion, the paper is hard to follow. There is no intuitive schematics that shows the main idea. The paper uses some terms that was no explicitly introduced (for example, CARD), but is essential to understand the paper. You need to go in the citations to get what the idea, which is not convenient.
2) Evaluation is only performed on the generated synthetic data, while it would be much more interesting to perturb and evaluate on real data from ImageNet.
3) No solution/defense was proposed to tackle the issue

**Questions:**

1) You mention in the introduction "Open-source text-to-image diffusion models, such as Stable Diffusion and DALL-E 2".  Is DALL-E 2 really open source?
2) If your propoposed attack is "Unrestricted Adversarial Example", how do you restrict it to have some certain $\ell_p$ norm? It's not mentioned in the text and confusing
3) How were parameters set? Did you do ablation studies on different choices of $w$, $a_1$, $a_2$, diffusion timestep?
4) What does $\delta=0.1$ line represent in Table 1? How is it different from previous lines?

---

### Official Review · Reviewer_ht9W · 2023-10-30

**Soundness:** 3 good
**Presentation:** 3 good
**Contribution:** 3 good
**Rating:** 5
**Confidence:** 4

**Summary:**

This paper proposes a novel adversarial attack method using diffusion models. This attack method can perform no-box attacks without relying on any training data. There are three steps to implement no-box adversarial attacks: 1) training a substitute model with diffusion models 2) fine-tuning the substitute model 3) generating adversarial attacks. Comprehensive experiments demonstrate that the proposed attack method can generate both no-box and black-box adversarial examples effectively and outperform the existing methods with higher attack success rate.

**Strengths:**

1. To perform no-box attacks, the authors utilize diffusion models to generate training dataset for substitute model conditioned on label or prompt label only. This process does not need the access to the target classifier’s training dataset or direct model queries.
2. Proposed attack method can reach sate of the art attack success rate for both no-box and black box setting compared with existing methods.
3. The authors successfully combine substitute models and diffusion generated adversarial examples (AdvDiff) which makes diffusion based adversarial attacks can be used in black-box and no-box attack setting.

**Weaknesses:**

1. Lack of evaluation with diffusion purification based defense method like DiffPure [1]. Since the adversarial examples are generated by UAE with a wrong adversarial guidance, could the unconditional purification remove such kind of adversarial guidance?
2. Potentially huge time cost for performing attacks. Generation of training dataset with diffusion models like Stable Diffusion or LDM seems to introduce high computational cost. Time complexity analysis is need compared with previous methods.
3. Lack of comparison with other diffusion based adversarial attack methods like DiffAttack [2].

[1] Weili Nie, Brandon Guo, Yujia Huang, Chaowei Xiao, Arash Vahdat, Anima Anandkumar. Diffusion Models for Adversarial Purification. arXiv preprint arXiv:2205.07460 (2022).

[2] Jianqi Chen, Hao Chen, Keyan Chen, Yilan Zhang, Zhengxia Zou, Zhenwei Shi. Diffusion Models for Imperceptible and Transferable Adversarial Attack. arXiv preprint arXiv:2305.08192 (2023).

**Questions:**

Please refer to the Weaknesses above.

**Details Of Ethics Concerns:**

No ethics concerns.

---

### Official Review · Reviewer_ZkvQ · 2023-10-30

**Soundness:** 2 fair
**Presentation:** 2 fair
**Contribution:** 2 fair
**Rating:** 3
**Confidence:** 3

**Summary:**

The paper delves into the use of diffusion models to craft human-imperceptible unrestricted adversarial examples. The methodology centers around a reverse generation process. In the experiments, this paper claims that noise sampling plays a pivotal role.

**Strengths:**

The paper focuses on the application of diffusion models for generating adversarial examples, a topic of significant relevance in the deep learning community.

**Weaknesses:**

1. The writing quality could be improved with a clearer problem motivation and a better explanation of some attack techniques. More intuition is needed in places.

2. This paper depends on re-trained diffusion models, which could potentially constrain the generalizability of the method. If there's a significant discrepancy between the data distribution of the target model and that of the diffusion model, the success rate of the attack will decrease.

3. The method enables the generating of vicious images that could spread misinformation or be used for malicious purposes. However, the authors do not discuss how to prevent misuse or unethical applications.

4. The paper lacks reproducibility statements or code references. In addition, the appendix should be included within the same document as the main content, but this paper appears to lack such integration.

**Questions:**

See the above.

**Details Of Ethics Concerns:**

However, the authors do not discuss how to prevent misuse or unethical applications.

---

### Official Review · Reviewer_UuiP · 2023-11-03

**Soundness:** 2 fair
**Presentation:** 2 fair
**Contribution:** 2 fair
**Rating:** 3
**Confidence:** 4

**Summary:**

The authors propose a method for no-box and black-box attacks using the Diffusion model, achieving promising results.

**Strengths:**

1. The authors are the first to propose generating data with the diffusion model for no-box attacks.
2. The experiments in the paper are relatively comprehensive.

**Weaknesses:**

1.  There are existing studies that generate data using the Diffusion model for Black-box attacks, such as [1]. Simply applying this idea to no-box attacks is not novel.

2.  In the experimental setup of the paper, the authors use the Stable-diffusion model to generate data for training the substitute models, and conduct no-box/black-box experiments using the ImageNet dataset. Is there any evidence to prove that there is no data from the ImageNet dataset in the training set of the stable-diffusion model used in this paper? If not, can it be claimed that such an experimental setup "does not use any training data from the target model"?

3. It is not surprising that the diffusion model can be used to generate adversarial examples. More importantly, it seems that the authors did not really test whether those adversarial examples can be easily defended (e.g., diffusion purification [2])

4. The overall organization of the paper and the use of symbols are somewhat chaotic, and the explanation of the formulas is insufficient. For example, the authors did not explain the meanings of most of the symbols in Eq. (3). Vectors are not represented using \mathbf{} as suggested, and the formatting of the subscripts in the symbols is also inconsistent.


[1] Shao, M., Meng, L., Qiao, Y., Zhang, L., & Zuo, W. (2023). Data-free Black-box Attack based on Diffusion Model. arXiv preprint arXiv:2307.12872.
[2] Nie, Weili, et al. "Diffusion models for adversarial purification." arXiv preprint arXiv:2205.07460 (2022).

**Questions:**

see above